# Notoginsenoside R1 Attenuates H/R Injury in H9c2 Cells by Maintaining Mitochondrial Homeostasis

**DOI:** 10.3390/cimb47010044

**Published:** 2025-01-10

**Authors:** Yuanbo Xu, Piao Wang, Ting Hu, Ke Ning, Yimin Bao

**Affiliations:** School of Integrative Medicine, Shanghai University of Traditional Chinese Medicine, Shanghai 201203, China; 22023022@shutcm.edu.cn (Y.X.);

**Keywords:** Notoginsenoside R1, hypoxia/reoxygenation, mitochondrial homeostasis, mitophagy

## Abstract

Mitochondrial homeostasis is crucial for maintaining cellular energy production and preventing oxidative stress, which is essential for overall cellular function and longevity. Mitochondrial damage and dysfunction often occur concomitantly in myocardial ischemia–reperfusion injury (MIRI). Notoginsenoside R1 (NGR1), a unique saponin from the traditional Chinese medicine Panax notoginseng, has been shown to alleviate MIRI in previous studies, though its precise mechanism remains unclear. This study aimed to elucidate the mechanisms of NGR1 in maintaining mitochondrial homeostasis in hypoxia/reoxygenation (H/R) H9c2 cells. The results showed that NGR1 pretreatment effectively increased cell survival rates post-H/R, reduced lactate dehydrogenase (LDH) leakage, and mitigated cell damage. Further investigation into mitochondria revealed that NGR1 alleviated mitochondrial structural damage, improved mitochondrial membrane permeability transition pore (mPTP) persistence, and prevented mitochondrial membrane potential (Δψm) depolarization. Additionally, NGR1 pretreatment enhanced ATP levels, increased the activity of mitochondrial respiratory chain complexes I–V after H/R, and reduced excessive mitochondrial reactive oxygen species (mitoROS) production, thereby protecting mitochondrial function. Further analysis indicated that NGR1 upregulated the expression of mitochondrial biogenesis-related proteins (PGC-1α, Nrf1, Nrf2) and mitochondrial fusion proteins (Opa1, Mfn1, Mfn2), while downregulating mitochondrial fission proteins (Fis1, Drp1) and reducing mitochondrial autophagy (mitophagy) levels, as well as the expression of mitophagy-related proteins (Pink1, Parkin, BNIP3) post-H/R. Therefore, this study showed that NGR1 can maintain mitochondrial homeostasis by regulating mitophagy, mitochondrial fission–fusion dynamics, and mitochondrial biogenesis, thereby alleviating H9c2 cell H/R injury and protecting cardiomyocytes.

## 1. Introduction

Myocardial ischemia–reperfusion injury (MIRI) is a significant concern in the treatment of ischemic cardiomyopathy [1]. Recent studies have identified various pathological mechanisms underlying MIRI, including excessive reactive oxygen species (ROS) production [2], Ferroptosis [3], inflammatory responses [4], and cell apoptosis [5]. Mitochondria, as the primary energy suppliers and key regulators of intracellular calcium homeostasis [6,7], are crucial in maintaining normal cardiac function and are increasingly recognized as a pivotal key in the pathogenesis of MIRI.

The heart, being the central organ of the circulatory system, relies on a continuous supply of energy. Consequently, cardiomyocytes contain a dense network of mitochondria that respond to energy demands through oxidative phosphorylation (OXPHOS) and the electron transport chain to produce ATP [8,9]. Mitochondrial integrity, encompassing both structure and function, is essential for cellular homeostasis. Disruptions such as reduced activity of mitochondrial respiratory chain complexes, decreased ATP synthesis, excessive ROS production, or irreversible opening of the mitochondrial membrane permeability transition pore (mPTP) can destabilize mitochondrial function, impair cardiomyocyte function, and exacerbate MIRI [10,11,12].

Mitochondrial homeostasis is maintained through both interrelated and independent processes, including mitochondrial autophagy (mitophagy), dynamics (fusion and fission) [13], and biogenesis [14]. Mitophagy is a cellular process in which damaged mitochondria are selectively degraded to maintain cellular health and energy balance. Mitochondrial dynamics, which involve mitochondrial fission and fusion [15], refer to the processes by which mitochondria combine (fusion) or divide (fission) to maintain their size, shape, and function. Together with mitophagy and mitochondrial biogenesis, they regulate mitochondrial structure, quantity and quality, physiological function, and programmed cell death, which are all critical for maintaining mitochondrial homeostasis [16]. Studies have shown that interventions enhancing mitochondrial fusion or inhibiting excessive fission and mitophagy can effectively mitigate MIRI [17,18].

Notoginsenoside R1 (NGR1) is a unique saponin component of Panax notoginseng, exhibiting pharmacological properties such as neuroprotection, inhibition of apoptosis and inflammation, and reduction in MIRI [19]. NGR1 has demonstrated protective effects on various tissues and organs [20,21,22,23]. Research indicates that NGR1 alleviates myocardial cell damage by upregulating miR-21 expression in response to oxygen and glucose deprivation or by suppressing the activity of TAK1 and JNK/p38 signaling to mitigate myocardial ischemia–reperfusion injury [24,25]. However, a systematic investigation into the mechanism by which NGR1 protects mitochondria against MIRI is still lacking.

Therefore, we hypothesize that NGR1 may exert its anti-MIRI effects by regulating mitochondrial homeostasis, preserving mitochondrial structure and function, and thus reducing cardiomyocyte damage.

## 2. Materials and Methods

### 2.1. Hypoxia/Reoxygenation (H/R) Model Establishment and Drug Administration

H9c2 cells (Cell Bank of the Chinese Academy of Sciences, Shanghai, China) were divided into the following groups: Control, Control + NGR1 (YuanYe Bio Co., Ltd., Shanghai, China, B21099), H/R, and H/R + NGR1. Prior to hypoxia-reoxygenation, drug pretreatment was performed: the Control + NGR1 and H/R + NGR1 groups were treated with a normal culture medium containing NGR1 (25 µM) for 2 h. After pretreatment, cells were subjected to hypoxia. The culture media of H/R + NGR1 and H/R groups were replaced with serum-free media with or without NGR1. The cell culture plates were placed in a hypoxia chamber with a gas mixture of 5% CO_2_ and 95% N_2_ for 6 h. After hypoxia, the culture media of H/R + NGR1 and H/R groups were replaced with normal media with or without NGR1, and the cells were incubated in a 37 °C incubator for reoxygenation for 2 h. The Control + NGR1 and Control groups were continuously cultured in normal media with or without NGR1 and kept in a 37 °C incubator throughout the experiment.

### 2.2. Cell Viability Assay

Cell viability was assessed using the CCK-8 reagent (Shanghai Biotech Co., Ltd., Shanghai, China, E6063350500). To assess the effect of different NGR1 concentrations on cell viability under normal or H/R conditions, cells were cultured in a 37 °C incubator for 10 h with media containing various concentrations of drugs. Alternatively, cells were pretreated with media containing different drug concentrations for 2 h, then replaced with serum-free and drug-containing media for hypoxia for 6 h, followed by reoxygenation with the same concentration of normal drug-containing media for 2 h. After these treatments, 10 µL of CCK-8 solution was added to each well and OD values at 450 nm were measured. Cell viability was calculated as follows: Cell viability (%) = [(Model group OD − Blank control OD)/(Normal control OD − Blank control OD)] × 100%.

### 2.3. Lactate Dehydrogenase (LDH) Detection

Cell injury was assessed by detecting changes in LDH levels in cell culture medium. The reagent (Nanjing Jiancheng Bioengineering Institute, Nanjing, China, A020-2-2) was mixed thoroughly and the absorbance was measured at 450 nm using a microplate reader. LDH content in the medium was calculated as follows: LDH content = [(Sample OD − Control OD)/(Standard OD − Blank OD)] × Standard concentration (0.2 mmol/L) × 1000.

### 2.4. Observation of Mitochondrial Ultrastructure

After hypoxia-reoxygenation, cells were collected, prepared into ultrathin sections and observed by transmission electron microscopy. The procedures included pre-fixation, rinsing, post-fixation, rinsing, dehydration, embedding, and sectioning. The sections were observed under a transmission electron microscope to examine mitochondrial morphology.

### 2.5. Detection of mPTP Opening

Cells were plated at a density of 2 × 10^5^ cells/well. Based on the experimental groups, cells underwent drug pretreatment and H/R modeling. The calcium green-cobalt fluorescence probe technique is a typical method for mPTP detection, utilizing the cell-permeable calcium green-AM fluorescent probe (GENMED, Batch No.: GMS10095.1V.A). The AM group is cleaved by intracellular esterases, generating highly fluorescent and polar calcium green. Under normal conditions, mPTP opens transiently, leading to rapid quenching of calcium green in the cytoplasm. In pathological states, mPTP remains persistently open, allowing Co^2+^ to enter the mitochondria and quench calcium green fluorescence, resulting in a gradual decrease in mitochondrial fluorescence intensity, which reflects the degree of mPTP opening. The intensity of green fluorescence was observed under a confocal laser microscope to determine the degree of opening of the mPTP, and the weakening of green fluorescence indicated that the mPTP was over-opened.

### 2.6. Mitochondrial Membrane Potential Measurement

JC-1 is a widely used fluorescent probe for assessing mitochondrial membrane potential (Δψm). Under normal physiological conditions, JC-1 accumulates in the mitochondrial matrix in an aggregated form, exhibiting red fluorescence; when Δψm depolarizes, JC-1 is released into the cytoplasm as monomers, displaying green fluorescence. Therefore, the shift in fluorescence color reflects changes in Δψm. Following the respective group treatments, cells underwent drug pre-treatment and H/R modeling. After hypoxia-reoxygenation, the staining working solution was added and incubated in the dark for 20 min. Subsequently, cells were washed twice with JC-1 staining buffer (Shanghai Beyotime Biotechnology Co., Ltd., Shanghai, China, C2006) and replenished with culture medium. Changes in Δψm were observed under a laser confocal microscope, and images were captured. The ratio of red to green fluorescence was used to quantify changes in Δψm, with a decreasing ratio indicating a decline in Δψm.

### 2.7. Mitochondrial Reactive Oxygen Species (MitoROS) Level Measurement

Following the respective treatments in different experimental groups, the culture medium was discarded, and cells were washed three times with pre-warmed HBSS. Subsequently, 200 µL of MitoTracker Green (Shanghai Yisheng Biotechnology Co., Ltd., Shanghai, China, 40742ES50) working solution was added, and the cells were incubated at 37 °C in the dark for 30 min. After removing the solution, cells were washed with HBSS and then incubated with 200 µL of MitoSOX Red (Shanghai Yisheng Biotechnology Co., Ltd., 40778ES50) working solution at 37 °C in the dark for 10 min. Finally, cells were washed with HBSS and stained with DAPI-containing anti-fade mounting medium. MitoROS levels were reflected by the average intensity of red fluorescence, while green fluorescence intensity was used as a proxy for mitochondrial quantity, observed under a laser confocal microscope.

### 2.8. Mitochondrial Respiratory Chain Complexes Activity Assay

After hypoxia-reoxygenation, 5 × 10^7^ cells were collected and resuspended in 1 mL of extraction buffer. The cells were homogenized using a homogenizer in an ice-water bath. The homogenate was centrifuged at 600× *g* for 10 min at 4 °C. The supernatant was collected and subjected to a second centrifugation at 11,000× *g* for 15 min at 4 °C to obtain the pellet. The pellet was resuspended in 400 µL of extraction buffer and mitochondria were disrupted using an ultrasonic homogenizer. The activities of respiratory chain complexes I to V were measured using corresponding assay kits (Beijing Solarbio Science & Technology Co., Ltd., Beijing, China, BC0515, BC3235, BC3245, BC0945, BC1445) following the manufacturer’s instructions. Absorbance values were recorded and used to calculate the activity of each complex.

### 2.9. Cellular ATP Level Measurement

After hypoxia-reoxygenation, the culture medium was discarded, and cells were lysed with an appropriate volume of lysis buffer (Shanghai Beyotime Biotechnology Co., Ltd., S0026). The lysate was centrifuged at 12,000× *g* at 4 °C and the supernatant was used for measurement. A standard ATP calibration curve and appropriate detection solutions were prepared. 100 µL of detection solution was added to a 96-well plate and incubated at room temperature for 3–5 min. Then, standard solutions and samples were added and mixed rapidly. The chemiluminescence was measured using a luminometer to determine the relative light units (RLU), and ATP content was quantified using the calibration curve.

### 2.10. Mitophagy Measurement

Mitophagy was assessed by dual staining of autophagosomes and lysosomes. Mtphagy Dye (Tongren Chemical, MD01) chemically binds to intracellular mitochondria, emitting weak fluorescence. When mitophagy occurs, autophagosomes are formed and fuse with lysosomes, leading to a decrease in pH and an acidic intracellular environment, which enhances the fluorescence intensity of Mtphagy Dye. Lyso Dye labels lysosomes and emits green fluorescence, allowing for the observation of autophagic lysosome formation when co-stained with Mtphagy Dye. Prior to hypoxia-reoxygenation, cells were incubated with an appropriate amount of Mtphagy working solution at 37 °C for 30 min. After washing with DMEM, H/R modeling was conducted, followed by incubation with an appropriate concentration of lysosomal working solution at 37 °C for another 30 min, and observations were made using a laser confocal microscope. The intensity of red fluorescence reflects changes in mitophagy.

### 2.11. Western Blotting of Mitochondrial-Associated Proteins

After hypoxia-reoxygenation, total cellular proteins or mitochondrial proteins were extracted using a mitochondrial isolation kit (Abcam, Cambridge, UK, ab110171). Protein quantification using the BCA method was performed, proteins were separated via SDS-PAGE, and transferred to PVDF membranes. They were blocked with BSA for 1 h, then incubated overnight at 4 °C with primary antibodies against Drp1 (CST, 5391T), Fis1 (Proteintech, 10956-1-AP), Opa1 (D6U6N) (CST, 80471S), Mfn1 (Abcam, ab57602), Mfn2 (Abcam, ab56889), Nrf1 (Abcam, ab175932), Nrf2 (Abcam, ab137550), PGC-1α (Abcam, ab54481), Pink1 (Abcam, ab23707), Parkin (CST, 4211s), and BNIP3 (Abcam, ab10433). After incubation, secondary antibodies, Anti-rabbit IgG HRP-linked (CST, 7074s) or Anti-mouse IgG HRP-linked (CST, 7076s), were added and incubated at room temperature for 1 h. The blots were developed using an enhanced chemiluminescence (ECL) reagent (Beyotime, 34096) and the band densities were quantified using ImageJ software(version 1.8.0; National Institutes of Health).

### 2.12. Statistical Analysis

The experimental data were analyzed using GraphPad Prism 8. All values are presented as mean ± standard deviation (x ± SD). Statistical comparisons between groups were performed using one-way ANOVA, with *p* < 0.05 considered statistically significant.

## 3. Results

### 3.1. Cell Damage and Survival

To confirm the protective effect of NGR1 on H/R cells, cell survival was measured on H9c2 cells treated with dose-gradient NGR1. Firstly, H9c2 cells were treated with gradient doses of NGR1 for 10 h, followed by an assessment of cell viability to determine the safe dosage of NGR1. In the dose range of 0~50 µM, there was no significant change in cell viability; however, at concentrations of 100 µM and above, cell viability significantly decreased (Figure 1A), indicating toxicity at these concentrations. Then, the experiment—in which cells pre-treated with different concentrations of NGR1 followed by H/R modeling—revealed a clear dose-dependent effect on cell viability. At concentrations of 12.5 µM, 25 µM, and 50 µM, NGR1 significantly enhanced cell survival compared to the H/R group (Figure 1B). Consequently, 25 µM was selected as the optimal dose for subsequent experiments. Additionally, content of LDH in the culture medium was tested. LDH release was significantly higher in the H/R group compared to the control group, but decreased in the H/R + NGR1 group (Figure 1C). These results suggest that NGR1 enhances cell viability and reduces cell damage, thereby protecting H9c2 cells from H/R injury.

### 3.2. Mitochondrial Structural Damage

Electron microscopy images revealed that H/R treatment caused significant damage to mitochondrial structure, including swelling, cristae disruption, and a reduction in matrix content. However, cells pretreated with NGR1 showed noticeable improvement in mitochondrial swelling, cristae breakage, and matrix content loss caused by H/R (Figure 2).

### 3.3. Mitochondrial Functional Damage

In addition to NGR1 attenuating the structural damage caused by H/R, its role in terms of mitochondrial functional damage has also been investigated. The periodic opening of mPTP is one of the conditions required for maintaining the Δψm, which is crucial for the normal functioning of oxidative phosphorylation. The results indicated a significant decrease in mPTP green fluorescence intensity and a reduction in the JC-1 red/green fluorescence ratio in H/R-treated cells, suggesting that H/R leads to sustained mPTP opening and Δψm depolarization. In contrast, cells pretreated with NGR1 exhibited a marked increase in mPTP green fluorescence intensity and JC-1 red/green fluorescence ratio compared to the H/R group (Figure 3). These suggest that NGR1 effectively alleviates sustained mPTP opening and Δψm depolarization caused by H/R.

Impairments in mitochondrial respiratory or energy metabolism functions are typically indicated by decreased activity of the electron transport chain (ETC) complexes or reduced ATP production, along with elevated ROS production. To understand the function change in aerobic respiration and ATP production in mitochondria after H/R, we investigated the activity of the complexes I–V that constitutes ETC and examined changes in ATP and mitoROS production. The results showed a significant decrease in ATP content following H/R treatment (Figure 4A), as well as a notable reduction in the activity of, ETC complexes I–V (Figure 4B–F). MitoSOX fluorescence staining revealed a marked increase in mitochondrial ROS production (Figure 4G). Cells pretreated with NGR1 exhibited a significant increase in ATP content (Figure 4A), improved the activity of ETC complexes I–V (Figure 4B–F), and reduced MitoSOX red fluorescence compared to the H/R group (Figure 4G–H). These results show that NGR1 effectively counteracts the explosive increase in mitoROS and mitigates electron transport chain damage, demonstrating its protection of mitochondrial respiratory and energy metabolic functions.

### 3.4. Mitochondrial Dynamics and Mitochondrial Biogenesis

The above results have demonstrated that NGR1 plays a positive role in maintaining mitochondrial structural and functional integrity. To explore the specific mechanisms by which NGR1 maintains mitochondrial homeostasis, we assessed indicators related to mitochondrial biogenesis and mitochondrial dynamics.

Mitochondrial biogenesis is crucial for generating new mitochondria to maintain their quantity and function. Cellular damage often suppresses mitochondrial biogenesis, leading to mitochondrial dysfunction. Results showed that the expression levels of PGC-1α, Nrf1, and Nrf2 were significantly reduced in the H/R group. In contrast, these levels were notably increased in the H/R + NGR1 group compared to the H/R group (Figure 5A–D). Besides mitochondrial biogenesis, mitochondrial dynamics also play a key role in maintaining mitochondrial quantity and function. Between the comparison of H/R and H/R + NGR1 group, H/R treatment led to a significant decrease in the expression levels of mitochondrial fusion proteins Opa1, Mfn1, and Mfn2 (Figure 5E–H), while the expression levels of mitochondrial fission proteins Drp1 and Fis1 were significantly increased (Figure 5I–K). In cells pretreated with NGR1, the expression levels of mitochondrial fusion proteins Opa1, Mfn1, and Mfn2 were significantly increased (Figure 5E–H), and the expression levels of mitochondrial fission proteins Drp1 and Fis1 were significantly reduced (Figure 5I–K). This suggests that NGR1 alleviates H/R-induced mitochondrial fusion inhibition and excessive fission.

### 3.5. Mitophagy

Mitophagy is a crucial mechanism for regulating mitochondrial quality and quantity, facilitating the degradation and clearance of damaged or excessive mitochondria. Mitophagy fluorescence staining provides a direct observation of this process. Results showed that in the H/R group, both red and yellow fluorescence intensities were significantly increased (Figure 6A,B), indicating a significant increase in the number of mitochondrial autophagosomes and autolysosomes. Similarly, the expression levels of mitophagy-related proteins Pink1, Parkin, and BNIP3 were notably increased in the H/R group (Figure 6C–F), suggesting a marked increase in mitophagy compared to control cells. Following NGR1 pre-treatment, there was a significant reduction in both red and yellow fluorescence intensities, and the expression levels of Pink1, Parkin, and BNIP3 proteins were also significantly decreased (Figure 6C–F), indicating a notable alleviation in mitophagy levels compared to H/R cells.

The main findings are as follows (Table 1).

## 4. Discussion

Notoginsenoside R1 (NGR1), a major bioactive component of Panax notoginseng, is a saponin with protopanaxatriol (PPT) as its aglycone. Recent investigations have revealed that NG-R1 undergoes novel metabolic transformations, including polyhydroxylation, glucuronidation, pentosylation, acetylation and amino acid conjugation [26]. Previous studies have indicated that NGR1 exerts a protective effect against myocardial injury [27]. For instance, NGR1 can attenuate ischemic heart failure by modulating MDM2/β arrestin2-mediated β2-adrenergic receptor ubiquitination [23], and it can also mitigate MIRI damage by modulating endoplasmic reticulum stress-related signaling pathways and oxidative stress [19,28]. On the H/R model of H9c2 cells, we observed that NGR1 can improve cell survival and has a protective effect on cardiomyocytes against H/R injury. It was also observed that NGR1 had an ameliorating effect on mitochondrial damage caused by H/R. Since mitochondria is the most abundant organelle in cardiomyocytes (its structural and functional integrity (mitochondrial homeostasis) is of crucial importance for maintaining the normal function of cardiomyocytes) we focused on the mechanism of action of NGR1 on mitochondria in this study.

The structural integrity of mitochondria is fundamental to their proper function [29]. Mitochondria are composed of a double membrane; the outer membrane is smooth and permeable, and the inner membrane is folded to form cristae, which surround a matrix containing enzymes and DNA and are responsible for energy production and metabolic regulation. The mitochondrial cristae are functional dynamic compartments, and the shape of the cristae is maintained by, among other things, mitochondrial fusion and division-associated proteins. Recently, it has been found that 1-deoxynojirimycin (DNJ) can facilitate mitochondrial function by targeting optic nerve atrophy protein 1 (OPA1) to promote its oligomerization, which leads to the reconstruction of mitochondrial cristae [30]. Our results also suggest that NGR1 attenuates H/R-induced mitochondrial structural damage, such as cristae break.

Mitochondria are vulnerable to external pathological processes, toxic substances, and endogenous and exogenous ROS, leading to mitochondrial membrane damage and excessive opening of the mPTP, which disrupts ion balance regulation [31]. During reperfusion, inflammatory factors, ROS, and calcium overload can cause sustained excessive opening of the mPTP, further leading to mitochondrial dysfunction [32,33]. In this study, we also observed that in H9c2 cells subjected to H/R injury, the fluorescence intensity of calcein significantly decreased, indicating that the mPTP loses its periodic opening and remains persistently open. NGR1 effectively ameliorated this phenomenon of continuous mPTP opening. The periodic opening of mPTP provides conditions for mitochondrial membrane potential (Δψm) generation, which is crucial for normal mitochondrial function. During MIRI, mitochondria exhibit vacuolation and swelling, with significant ROS accumulation exacerbating oxidative stress [34]. This also leads to mitochondrial membrane potential depolarization, further triggering apoptosis via the mitochondrial pathway [35]. In conclusion, the protective effects of NGR1 on mPTP and mitochondrial membrane potential in H/R cells suggest that it can improve the function of mitochondria.

Mitochondria are central to cellular respiration and energy metabolism. Oxidative phosphorylation is the process by which electrons are transferred through the respiratory chain to generate a potential difference and catalyze ATP synthesis [36]. The mitochondrial respiratory chain involves respiratory chain complexes I–V, and their enzyme activity is closely related to mitochondrial energy metabolism and respiratory function. Mitochondrial dysfunction may be closely related to impaired enzyme activity. Additionally, mitochondrial respiration is a significant source of intracellular ROS. Impaired mitochondrial respiration leads to ROS accumulation and exacerbates oxidative stress [37]. ROS accumulation can also inhibit mitochondrial oxidative phosphorylation, suppress mitochondrial respiration, and reduce ATP production, ultimately resulting in myocardial cell or tissue damage or death [38]. Our study found that H9c2 cells subjected to H/R showed a significant reduction in the activity of mitochondrial respiratory chain complexes, accompanied by mitoROS accumulation and a marked decrease in ATP production. Pre-treatment with NGR1 notably increased the activity of mitochondrial respiratory chain complexes, alleviated mitoROS accumulation, and significantly enhanced ATP production. These results suggest that NGR1 provides substantial protection to mitochondrial function in H9c2 cells following H/R.

We then further explored the mechanisms by which NGR1 ameliorates mitochondrial structural and functional impairment. Mitochondrial biogenesis, mitochondrial dynamics, and mitophagy are altered when mitochondria are subjected to adverse stimuli, which have a buffering effect on the maintenance of mitochondrial structural and functional integrity.

Mitochondrial biogenesis is primarily regulated by PGC-1α, which controls downstream factors that produce new mitochondria and indirectly affects mitochondrial respiration and energy metabolism [39]. Mitochondrial biogenesis is regulated by PGC-1α, which upon activation through phosphorylation or de-acetylation, subsequently activates Nrf1 and Nrf2, leading to the expression of mitochondrial transcription factor A (Tfam) [40,41]. Studies have shown that ischemia–reperfusion injury reduces PGC-1α expression [42]. Our results similarly show that H/R reduced the expression levels of PGC-1α, Nrf1, and Nrf2; however, NGR1 significantly increased these protein levels. This indicates that NGR1 effectively improves suppressed mitochondrial biogenesis and protects mitochondrial function.

Mitochondrial dynamics, including fusion and fission, help maintain mitochondrial quantity and function and are thought to be a possible new target for the treatment of atherosclerosis [43]. Mitochondrial fusion allows damaged or fragmented mitochondria to repair themselves, while fission helps segregate damaged parts for degradation through autophagy [44]. Excessive mitochondrial fission or inhibited fusion can lead to irreversible damage in myocardial cells. In our study, after H/R, the fission proteins Drp1 and Fis1 were markedly increased, while fusion proteins Opa1, Mfn1, and Mfn2 were significantly reduced, indicating increased mitochondrial fission and suppressed fusion. NGR1 significantly improved excessive mitochondrial fission and suppressed fusion.

Under physiological conditions, fragmented mitochondria formed by fission or other damaged mitochondria can be degraded through mitophagy. One of the classical pathways mediating mitophagy is the Pink1/Parkin pathway [45], and receptors like BNIP3 on the mitochondrial inner membrane can also directly interact with LC3 to mediate mitophagy [46]. Once damage exceeds the mitophagy threshold, excessive mitophagy can exacerbate damage, creating a vicious cycle. In our study, fluorescence staining and mitophagy-related protein expression assays showed significantly increased mitophagy and elevated Pink1, Parkin, and BNIP3 protein levels in H/R cells, indicating enhanced mitophagy. After NGR1 pre-treatment, mitophagy was significantly reduced, and the expressions of Pink1, Parkin, and BNIP3 proteins decreased, suggesting that NGR1 effectively inhibits excessive mitophagy (Figure 7). One study limitation that needs to be pointed out is that the experimental model used (H9c2 cells), which cannot fully represent the complexity of a living being. It also needs to be underlined that further in vivo studies or clinical trials are required [47].

Compared to conventional cardiovascular drugs, such as ACE inhibitors, β-blockers, and antiplatelet agents, NGR1 offers the advantage of multi-target actions that more effectively protect myocardial cells and improve heart function [26,48,49]. Furthermore, NGR1 has a relatively low incidence of side effects and shows promising long-term safety profiles. However, we acknowledge that clinical data on NGR1 remains limited at this stage. Further clinical validation, as well as pharmacokinetic studies, are necessary to fully establish its therapeutic potential. Future research could explore strategies such as targeted drug delivery systems and nanotechnology to enhance the bioavailability and targeting specificity of NGR1, thereby increasing its effectiveness in myocardial cells.

## 5. Conclusions

In conclusion, our study demonstrates that NGR1 can maintain mitochondrial homeostasis and alleviate H/R injury in H9c2 cells by inhibiting mitophagy, promoting mitochondrial biogenesis, and regulating mitochondrial dynamics.

## Figures and Tables

**Figure 1 cimb-47-00044-f001:**
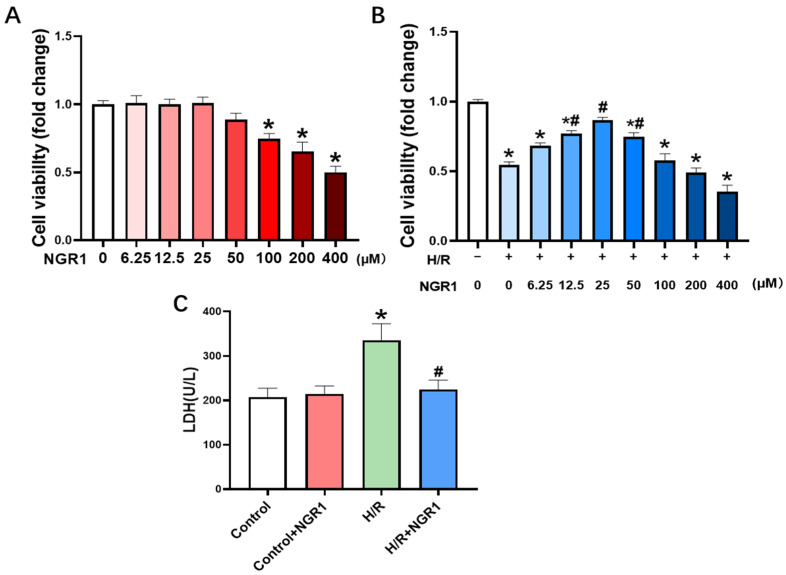
NGR1 increases cell survival and attenuates cell damage after H/R. (**A**) Effect of different concentrations of NGR1 on the survival of H9c2 cells, *n* = 8. (**B**) Effect of different concentrations of NGR1 on the survival of H9c2 cells after H/R, *n* = 8. (**C**) Effect of NGR1 on the release of LDH from H9c2 cells after H/R, *n* = 6. * *p* < 0.05 vs. Control group, # *p* < 0.05 vs. H/R group.

**Figure 2 cimb-47-00044-f002:**
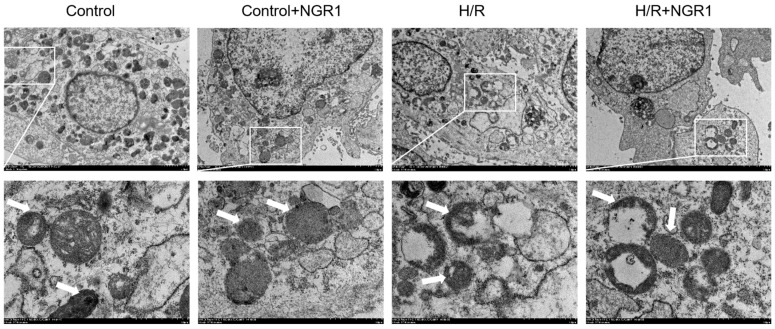
NGR1 attenuates mitochondrial structural damage in H9c2 cells after H/R. The white boxed portion is enlarged in the figure below, and the portion indicated by the arrow undergoes mitochondrial ultrastructural changes. bar = 2 µm.

**Figure 3 cimb-47-00044-f003:**
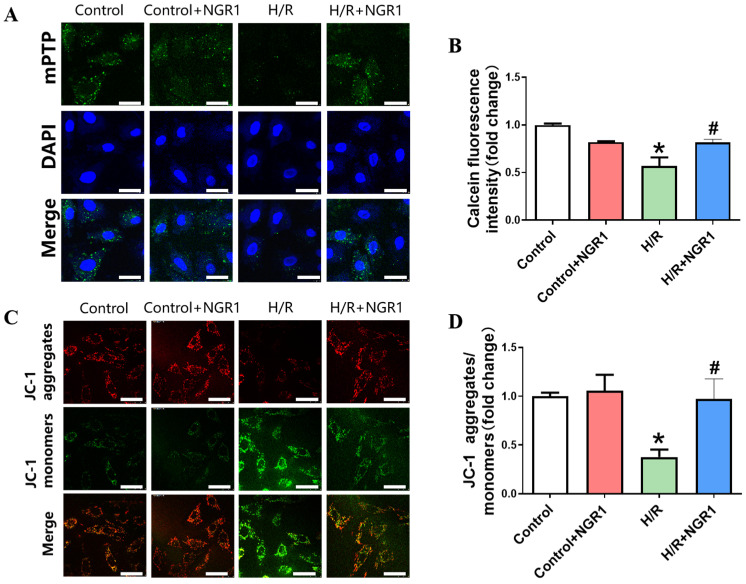
NGR1 inhibits mPTP persistent opening and Δψm depolarization. (**A**) Calcein staining results of H9c2 cells in each group. (**B**) Calcein fluorescence intensity statistics results. (**C**) JC-1 staining results of H9c2 cells in each group. (**D**) Statistical results of JC-1 aggregates/monomers. * *p* < 0.05 vs. Control, # *p* < 0.05 vs. H/R, *n* = 5, bar = 50 µm.

**Figure 4 cimb-47-00044-f004:**
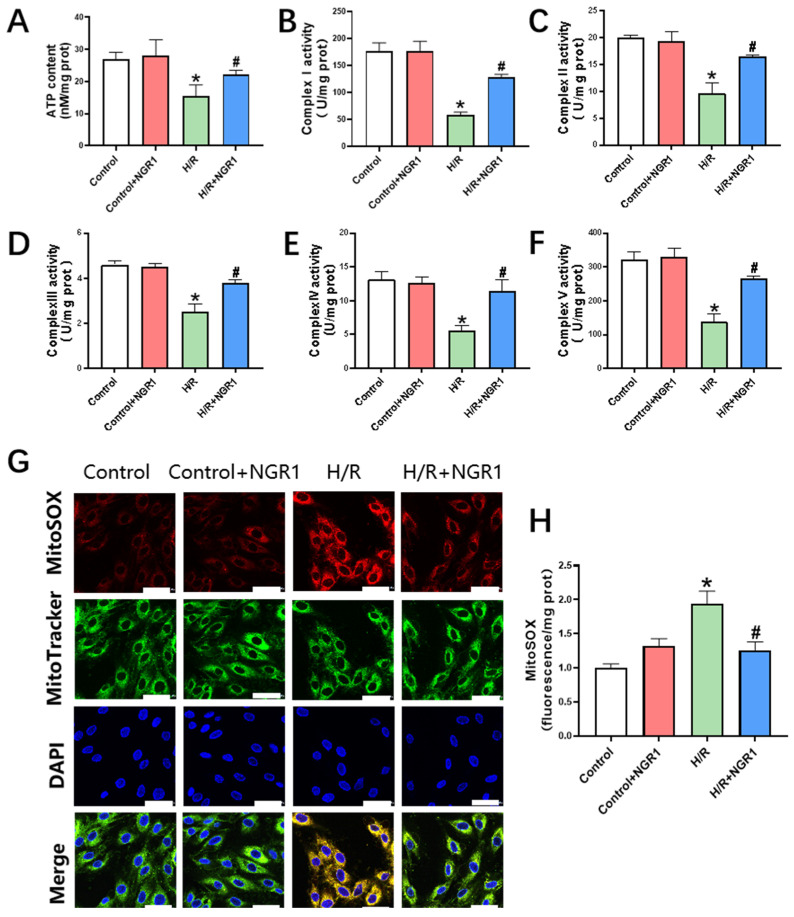
NGR1 increases ATP content and improves the impaired respiratory function. (**A**) ATP content of H9c2 cells in different groups, *n* = 6. (**B**–**F**) Activities of mitochondrial respiratory chain complex I, II, III, IV, V in H9c2 cells in different groups, *n* = 5~6. (**G**,**H**) MitoSOX fluorescent staining images and fluorescence intensity statistics. Results, *n* = 5, bar = 50 µm. * *p* < 0.05 vs. Control; # *p* < 0.05 vs. H/R.

**Figure 5 cimb-47-00044-f005:**
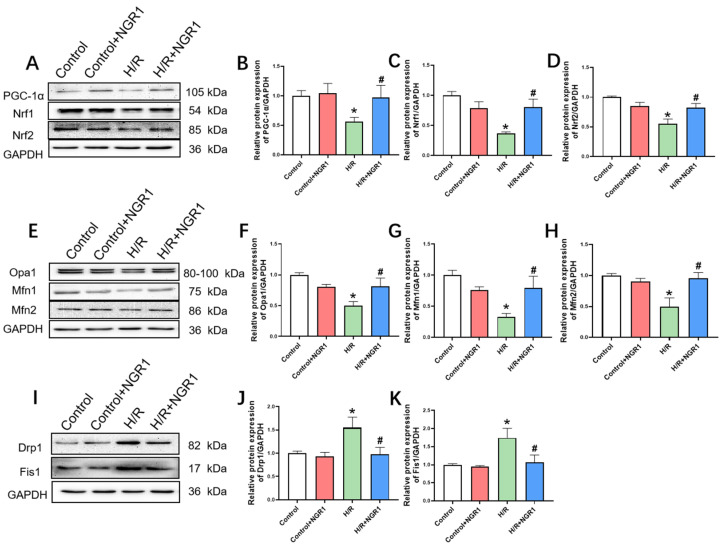
NGR1 increases mitochondrial biogenesis and improves mitochondrial dynamics. (**A**–**D**) Typical bands and statistical results of the expression of PGC-1α, Nrf1, Nrf2 proteins, *n* = 6. (**E**–**H**) Typical bands and statistical results of the expression of mitochondrial fusion proteins Opa1, Mfn1, and Mfn2, in which F, H: *n* = 6, and G: *n* = 4. (**I**–**K**) Mitochondrial fission protein Drp1, Fis1 expression typical bands and statistical results, *n* = 8. * *p* < 0.05 vs. Control; # *p* < 0.05 vs. H/R.

**Figure 6 cimb-47-00044-f006:**
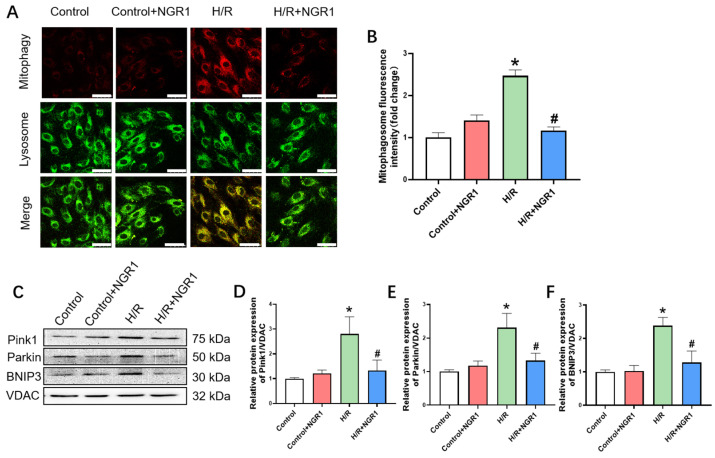
NGR1 can inhibit excessive mitophagy after H/R. (**A**) Mitochondrial autophagosome and lysosome fluorescence double staining results. bar = 50 µm. (**B**) Statistical results of fluorescence intensity of mitophagy, *n* = 3. (**C**–**F**) Typical bands of mitophagy-related proteins Pink1, Parkin, and BNIP3 expression and statistical results. * *p* < 0.05 vs. Control; # *p* < 0.05 vs. H/R, *n* = 6.

**Figure 7 cimb-47-00044-f007:**
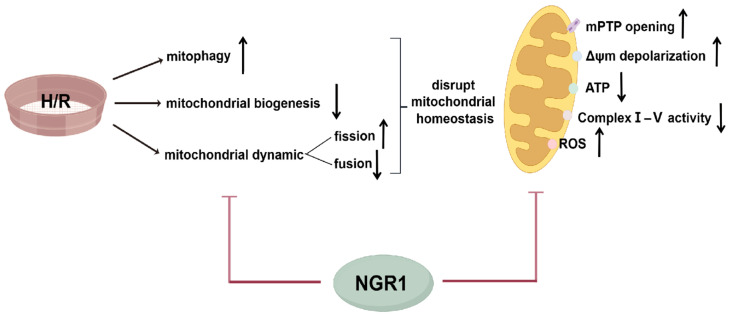
Mechanisms by which NGR1 maintain mitochondrial homeostasis after H/R. NGR1 maintain mitochondrial homeostasis after H/R through inhibiting mitophagy, promoting mitochondrial biogenesis and modulating mitochondrial dynamics, thereby mitigating mitochondrial damage, including reducing mitoROS production, inhibiting the mPTP persistence opening and inhibiting Δψm depolarization, as well as increasing the production of ATP and respiratory chain complexes enzyme activity.

**Table 1 cimb-47-00044-t001:** Summary of major findings.

Observations			H/R	NGR1
Cell damage and viability	Cell viability	OD value	↓	↑
	Cell damage	LDH	↑	↓
Mitochondrial structure and function	Structure	Swelling, etc.	↑	↓
	mPTP	Green fluorescence intensity	↓	↑
	Δψm	JC-1 aggregates/monomers	↓	↑
	Energy synthesis	ATP	↓	↑
	ETC complexes I-V	Activities	↓	↑
	Oxidative stress	ROS	↑	↓
Mechanisms of mitochondrial homeostasis	Mitochondrial Biogenesis	PGC-1α, Nrf1, and Nrf2	↓	↑
	Mitochondrial Dynamics	Opa1, Mfn1, and Mfn2	↓	↑
		Drp1 and Fis1	↑	↓
	Mitophagy	Fluorescence intensities	↑	↓
		Pink1, Parkin, and BNIP3	↑	↓

## Data Availability

The data used to support the findings of this study are included within the article.

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
