# Peer review of "Notoginsenoside R1 Attenuates H/R Injury in H9c2 Cells by Maintaining Mitochondrial Homeostasis"

_cimb, 2025, doi:10.3390/cimb47010044_

Round 1

Reviewer 1 Report

Comments and Suggestions for Authors

This study was concerned with the investigation of the mechanisms by which Notoginsenoside R1 (NGR1) maintains mitochondrial homeostasis in hypoxia/reoxygenation (H/R) H9c2 cells. The results showed that NGR1 pretreatment effectively increased cell survival rates post-H/R, reduced lactate dehydrogenase leakage, mitigated cell damage, alleviated mitochondrial structural damage and improved mitochondrial function. The mechanism of NGP1 activity was associated with NGR1-mediated upregulation of expression of mitochondrial biogenesis-related proteins and mitochondrial fusion proteins and downregulation of mitochondrial fission protein expression and reduction of mitochondrial autophagy levels. It was concluded that the protective effect of NGR1 on cardiomyocytes was associated with its ability to maintain mitochondrial homeostasis, which was associated with its regulatory effect on mitophagy, mitochondrial fission-fusion dynamics, and mitochondrial biogenesis.

Major points:

1. All the control cells should also be cultivated in serum-free medium from 6 hours but without exposure to the hypoxia condition. This will verify that the reduced cell viability in H/R cells is due to hypoxia only but not due to the serum starvation and medium change. 

2. Please clarify if the H9c2 rat myoblasts were used in the study.

3. Figure 3B, 3D, 4H, 5B-5D, 5F-5H, 5J, 5K, 6B, 6D-6F, no error bar for the Control. Did it mean that the Control group had a standard deviation (SD) value of 0? If the SD value were 0 for the control group, the t-test would not be needed.  

Reviewer 2 Report

Comments and Suggestions for Authors

General comment: The manuscript presented by Xu et al. is overall well written and provides some very interesting results concerning the beneficial effects of NGR1 administration to overcome major alterations in H9c2 cells, mostly associated with mitochondria, promoted by hypoxia followed by reoxygenation (“reperfusion”).   

Major comments:

i)                The authors use H9c2 cells in the study. They should provide a rational for their use instead of primary cardiomyocytes and how effectively they could mimic their functions in MIRI.

ii)              The authors refer to H/R when referring to the cells. The term reperfusion is conventionally applied to organ perfusion (e.g., heart reperfusion) and not to cells in culture. Could they comment on this? Reperfusion in this context refers mostly to allowing the presence of oxygen again in the medium instead of a true reperfusion process.

iii)             The methods section is much based on application of kits. Is it not possible to derive similar data in some of the experiments using more intuitive methodological approaches besides kits. With this I do not want to suggest that the described approaches are incorrect, however it sounds awkward that the majority of the methodological approaches used are based in kits utilization.

iv)             The authors need to provide in their discussion section some kind of reasoning for such holistic effects of NRF1. For example, which structural characteristics of NRF1 might be involved in all the detected rescuing properties. The multi-target action of NRF1 must certainly be associated with structural specificities of the molecule.

Minor comments (some of many):

i)                Page 1, line 27: “as well as the expression of mitophagy -related proteins (Pink1, Parkin, BNIP3) decreased post-H/R.”; this segment of the sentence needs correction.

ii)              The introduction section starts with the abbreviation MIRI; although in the abstract the authors provide the meaning of such abbreviation, they should also provide it in the beginning of the manuscript itself. Thus, I suggest that for its first usage in introduction the authors should provide its full name. The same for other abbreviations which meaning is provided in abstract but not in the manuscript body itself.

iii)             Page 2 line 51: instead of and just use comma; the sentence will sound better.

iv)             Page 2 line 53: “quanlity”; correct to quality.

v)              Page 2 line 59: the authors refer to reference 19 to mention the pharmacological properties of NGR1; this reference is a mini-review. I suggest to the authors that they should instead provide original research on NGR1 on each of the pharmacological properties.

vi)             Page 2 line 64: “affects on mitochondria to against”; this segment of the sentence needs correction.

vii)            Page 32 line 73: “group”; use groups instead of group; two groups are being considered ion the sentence.

viii)           Page 3 lines 97, 103, 139, 149, 178, 185: “After modelling” of “Prior to modeling”; what do the authors mean by this?

ix)             Page 3 line 129: “Following experimental grouping”; what exactly is meant by this expression.

x)               Page 3 lines 138, 180: “H/R modeling according to experimental groups”; there must be some other way to refer to the experimental treatment being given to the cells.

xi)             Page 4 line 148: “complex I-V”; the authors refer to 5 complexes, I-V, thus the term complex should be corrected to complexes. The same for Page 6, line 246, Page 12 line 395.

xii)            Page 4 line 163: “at 12,000 g for 4℃”; for 4°C? what is exactly this? Do you mean at 4°C?

xiii)           Page 5 lines 194-195: “Develop using ECL reagent (Beyotime, 34096) and analyze 194 band density values with Image J software.”; what a strange way to refer to methods and methodologies. Same for Page 5 lines 197-199: “Analyze experimental data using GraphPad Prism 8. Data are expressed as mean ± standard deviation (x ± SD). Compare groups using one-way ANOVA, with p< 0.05 considered statistically significant.”

xiv)           Page 5 line 213: “LDH is a marker of myocardial injury”; this was already mentioned in materials and methods section and is widely known; no need for repetition.

xv)            Page 6 line 226: “mitochondrial swelling, cristae breakage, and matrix content loss”; does the loos term applies to mitochondrial swelling, cristae breakage and matrix content? If that is the case than instead o loss should be written losses!!

xvi)           Page 8 line 273: “dynamic”; should be corrected to dynamics.

xvii)          Page 8 lines 283-286: the authors need to refer that the results described in the sentences report to the comparison between H/R and H/R+NGR1.

xviii)         Page 10 line 320: “Since mitochondria, as the most”; correct to Since mitochondria is the most. As is makes no sense in the context of the rest of the text in the sentence.

Reviewer 3 Report

Comments and Suggestions for Authors

The present study investigates the role of Notoginsenoside R1 (NGR1), a saponin extracted from the plant Panax notoginseng, in protecting H9c2 cells from hypoxia/reoxygenation (H/R) injury. The present study demonstrates that NGR1 enhances cell viability and reduces cellular damage while maintaining mitochondrial function by maintaining mitochondrial homeostasis. More specifically, NGR1 regulates mitophagy, mitochondrial fusion and fission dynamics, and mitochondrial biogenesis, protecting against oxidative stress, loss of membrane dynamics, and decreased ATP synthesis.

From my perspective, this article is of significant importance because mitochondrial homeostasis is extremely important in dealing with cardiomyocyte injury induced by hypoxia and the subsequent reoxygenation injury. NGR1 seems to be a probable therapeutic approach for the treatment of myocardial ischemia-reperfusion injury.

In my opinion the article could be published if some major issues are addressed:

1.      Introduction: A few of the technical terms, such as mitophagy and mitochondrial fusion and fission dynamics, are treatable with straightforward explanations, thereby making such concepts comprehensible to nonspecialists in that specific field.

2.      Statistical results: There is less use of frequent statistical jargon—e.g., p<0.05—and more consideration of a summary table that summarizes the main findings in order to facilitate the understanding of the reader.

3.      Results: Some of the results would best be presented in a much more summarized manner. Such results could be put in sub-sections with, for example, such sub-headings as stating main findings:

• Cell damage and survival

• Mitochondrial function

• Mechanisms of homeostasis

4.      Discussion: The study results and their implications for therapeutic use could be more clearly separated. This would increase the understanding of what the research adds.

5.      Summary tables: Making summary tables of the results of experiments by category might make them easier to analyze comparatively and review (for example, LDH, ATP, mitoROS).

6.      Clinical implications: It would be worthy to delve deeply into how the results of this research can be applied to bring developments in pharmacological therapy to address cardiac disorders.

7.      Study limitations: A clear statement of the limitations of the study can increase the degree of objectivity. For instance, it has to be pointed out that the experimental model used (H9c2 cells) cannot fully represent the complexity of a living being. It also needs to be underlined that further in vivo studies or clinical trials are required.

8.      Comparisons to the existing therapies: It might be quite interesting to learn how NGR1 would compare to other existing pharmaceuticals or substances that interact with mitochondrial homeostasis.

Round 2

Reviewer 1 Report

Comments and Suggestions for Authors

In response to my concerns regarding the missing SD for the controls in Figure 3B, 3D, 4H, 5B-5D, 5F-5H, 5J, 5K, 6B, 6D-6F, the authors mentioned that the control group was normalized to 100%. If that was the case, it should indicate in the label of y-axis that the data represent "fold change relative to the control". It was not clear whether the normalized data or data without normalization were used for the one-way ANOVA analysis. Typically, the values used to generate the bar graph should be the same as those used in the statistical analysis. It does not make sense if the statistical results presented on a graph are not based on the data used to generate the graph. In this regard, if SD equals zero for the control group after normalization, performing a meaningful one-way ANOVA would be impossible. To solve the problem of zero SD, each value in a set of data points should be divided by the mean value of the control group and in doing so, the control group should have a mean value of 1 (or 100%) and a SD value. Please revise the figures and repeat the statistically analysis accordingly.  

Reviewer 2 Report

Comments and Suggestions for Authors

I am pleased with the detailed answers provided by the authors to each of the raised questions. 

Just a minor comment on a sentence that the authors added to the discussion section: Page 13 lines 410-412 of revised version. The sentence "Of course the Study limitations has to be pointed out that the experimental model used (H9c2 cells) cannot fully represent the complexity of a living being." The first segment of the sentence needs correction; maybe something like: “One study limitation that needs to be pointed out is that the experimental cell model used (H9c2 cells) cannot fully represent the complexity of a living being.

Reviewer 3 Report

Comments and Suggestions for Authors

The authors have revised the manuscript in accordance with the suggestions provided. I wholeheartedly recommend the publication of this work.

Round 3

Reviewer 1 Report

Comments and Suggestions for Authors

The authors have addressed all of my concerns.